# Clinically compliant spatial and temporal imaging of chimeric antigen receptor T-cells

Nia Emami-Shahri[1], Julie Foster[2], Roxana Kashani[2], Patrycja Gazinska[3], Celia Cook[1], Jane Sosabowski[2], John Maher[4,5,6] & Sophie Papa [1,7]

The unprecedented efficacy of chimeric antigen receptor (CAR) T-cell immunotherapy of CD19[+] B-cell malignancy has established a new therapeutic pillar of hematology–oncology. Nonetheless, formidable challenges remain for the attainment of comparable success in patients with solid tumors. To accelerate progress and rapidly characterize emerging toxicities, systems that permit the repeated and non-invasive assessment of CAR T-cell biodistribution would be invaluable. An ideal solution would entail the use of a non-immunogenic reporter that mediates specific uptake of an inexpensive, non-toxic and clinically established imaging tracer by CAR T cells. Here we show the utility of the human sodium iodide symporter (hNIS) for the temporal and spatial monitoring of CAR T-cell behavior in a cancer-bearing host. This system provides a clinically compliant toolkit for high-resolution serial imaging of CAR T cells in vivo, addressing a fundamental unmet need for future clinical development in the field.

[1] ImmunoEngineering Group, King's College London, Division of Cancer Studies, 3rd Floor Bermondsey Wing, King's Health Partners Integrated Cancer Centre, Great Maze Pond, Guy's Hospital, London SE1 9RT, UK. [2] Centre for Molecular Oncology, Barts Cancer Institute, Queen Mary University of London, Charterhouse Square, London EC1M 6BQ, UK. [3] Breast Cancer Now, Division of Cancer Studies, Guy's Cancer Centre, Great Maze Pond, London SE1 9RT, UK. [4] CAR Mechanics Group, King's College London, Division of Cancer Studies, 3rd Floor Bermondsey Wing, King's Health Partners Integrated Cancer Centre, Great Maze Pond, Guy's Hospital, London SE1 9RT, UK. [5] Department of Clinical Immunology and Allergy, King's College Hospital NHS Foundation Trust, London SE5 9RS, UK. [6] Department of Immunology, Eastbourne Hospital, King's Drive, Eastbourne BN21 2UD, UK. [7] Department of Medical Oncology, Guy's and St Thomas' NHS Foundation Trust, Great Maze Pond, London SE1 9RT, UK. These authors jointly supervised this work: Jane Sosabowski, John Maher, Sophie Papa. Correspondence and requests for materials should be addressed to S.P. (email: sophie.papa@kcl.ac.uk)

Chimeric antigen receptors are genetically delivered fusion molecules that couple the binding of a native tumor-associated cell surface target to delivery of a bespoke T-cell activating signal[1,2]. Efficient disease control by CAR T cells has been demonstrated in pre-clinical models representative of a broad range of cancer types[1,3–9]. Unprecedented clinical impact has been achieved in patients with refractory B-cell malignancies[10–14], with complete remissions in heavily pre-treated patients highlighting the truly groundbreaking nature of this advance. Some patients do not gain benefit from CD19-targeted CAR T-cell therapy, exhibiting primary resistance. Appreciation of the frequency of disease relapse, either with CD19-positive or CD19-negative disease, is growing as more clinical experience is gained in CD19 expressing hematological malignancies[13]. Moreover, patients may endure severe side effects due to cytokine release syndrome (CRS), neurotoxicity, or on-target off-tumor toxicity that is frequently unanticipated prior to first in man evaluation[15,16]. The next anticipated breakthrough is the demonstration of meaningful efficacy of CAR T-cell immunotherapy in patients with solid tumors. This will require that CAR T cells migrate to, penetrate and then persist in an active state within a tumor microenvironment that is profoundly immunosuppressive[8,11,12,14,17–21]. Given these considerations, pre-clinical and early clinical development of novel CAR T-cell therapies would be greatly facilitated if we could undertake repeated and reliable tracking of these cells after their infusion in animal studies and patients. An ideal approach would be non-invasive, cost-efficient and equally compatible with both small animal and clinical imaging modalities[22].

Single-photon emission-computed tomography (SPECT/CT) tracking of indium-111 ($^{111}$In) labeled CAR T cells provides a brief snapshot of the fate of adoptively transferred cells in vivo[23,24]. This approach gives good image resolution but is hampered by key limitations. First, due to isotope decay, signal is lost within 96 h. Second, labeling is agnostic to CAR expression by T cells. Thirdly, passive labeling does not report on cellular proliferation following adoptive transfer as signal is not maintained in daughter cells and, finally, activity may be mis-registered through phagocytosis of dying labeled cells. Co-expression of a CAR and a reporter gene within the same cell can overcome these limitations. Proof of concept was first demonstrated using the herpes simplex virus thymidine kinase 1 (HSV1tk) reporter, co-expressed with a CAR and luciferase reporter. This system enabled the serial imaging of CAR T cells using both positron emission tomography (PET) and bioluminescence imaging (BLI)[25]. More recently, PET imaging of HSV1tk$^+$ CAR T cells has been achieved in patients with high-grade glioma[26]. However, HSV1tk is a viral protein which is immunogenic in man, favoring immune-mediated recognition and elimination of HSV1tk transduced T cells[27]. Use of a human reporter gene, such as norepinephrine receptor or hNIS, would overcome this concern[28,29]. As yet however, neither has been successfully adapted to achieve real-time imaging of CAR T cells. The somatostatin receptor type 2 (SSTR2) has been recently co-expressed with an intracellular adhesion molecule-1 directed CAR and imaged by PET-CT with gallium-68-labeled octreotide analog ($^{68}$Ga-DOTATOC)[30,31]. However, this approach has two limitations. Firstly, the SSTR2 receptor internalizes on interaction with its substrates, risking poor sensitivity, especially at lower reporter gene expressing cell density[32]. Secondly, SSTR2 is expressed on T cells and other immune cells[33], accounting for the ability of octreotide analogs and their radiolabeled derivatives to inhibit T-cell function[34]. This is clearly undesirable for a broadly applicable strategy to image therapeutic T-cell products in cancer patients.

The hNIS gene is localized on chromosome 19p12-13.2 and encodes a 643 amino acid glycoprotein with a molecular mass of approximately 70–90 kDa. Cloning and sequencing of the human NIS gene was completed twenty years ago[35]. The hNIS is a member of the sodium-dependent transporter family and is a membrane spanning protein with 13 putative transmembrane domains, an extracellular amino-terminus and an intracellular carboxyl-terminus. The symporter co-transports two sodium ions with one iodide ion, capitalizing on the electrochemical potential of the sodium gradient generated by the sodium-potassium ATPase (Na$^+$/K$^+$-ATPase) pump of the plasma membrane, to rapidly concentrate inorganic iodide within the thyroid gland. The Na$^+$/K$^+$-ATPase can be blocked by ouabain as well as by the competitive inhibitors thiocyanate and perchlorate[36]. This symporter is naturally found in stomach, salivary glands, lactating breast and in some thyroid cancers[37]. Expression of hNIS may also be detected in some non-thyroid human cancers although, in these circumstances, it is generally located intracellularly, rather than on the cell surface[38–40]. The hNIS symporter has long been exploited in clinical medicine owing to its ability to mediate uptake by the thyroid gland of a variety of radioisotopes, both for diagnostic and therapeutic applications. In the context of CAR T-cell imaging, hNIS represents an ideal candidate reporter gene since it is non-immunogenic, is not internalized upon substrate uptake, and is only functional in living cells[41]. Furthermore, hNIS is compatible with cheap and widely used clinical radiotracers, notably technetium-99m pertechnetate ($^{99m}$TcO$_4^-$), which is routinely administered to patients for red blood cell imaging. Genetically delivered hNIS has been incorporated in oncolytic virus-based clinical trials[42] and in pre-clinical studies of leukocyte imaging[29,43,44], providing important additional experience. Cognizant of these attributes, we set out to evaluate the use of hNIS as an imaging reporter for CAR T-cell immunotherapy. In a model of CAR T-cell prostate cancer immunotherapy the inclusion of hNIS in the retroviral vector for T-cell transduction was validated in vitro and in vivo. Radiotracer kinetics were defined in transduced T cells. Finally, SPECT/CT imaging with $^{99m}$TcO$_4^-$ radiotracer over the course of CAR T-cell immunotherapy of high and low tumor burden bearing hosts demonstrates the utility of this approach for CAR T-cell imaging.

## Results

**Generation of a robust in vivo model of prostate cancer**. An essential requirement in this quest is an in vivo model that demonstrates robust and reliable anti-tumor activity of human CAR T cells. P28ζ is a second generation CAR that enables human T cells to elicit potent and precisely targeted anti-tumor activity against prostate specific membrane antigen (PSMA)-expressing prostate cancer cells[2] and derived tumors[45]. We developed a highly reproducible model to demonstrate in vivo function of this CAR using PC3-LN3 (PL) prostate cancer cells. Since PL tumor cells lack PSMA, they were engineered to co-express firefly luciferase and tdTomato-red-fluorescent protein alone (PL), or together with PSMA (PLP; Fig. 1a). Ectopic expression of PSMA did not alter growth kinetics in vitro (Fig. 1b), but in vivo after subcutaneous (s.c.) inoculation the expression of PSMA leads to a moderately more aggressive phenotype (Fig. 1c). Importantly tumor cells did not express hNIS (Fig. 1d) and consequently did not exhibit in vitro $^{99m}$TcO$_4^-$ uptake (Fig. 1e). When propagated as s.c. xenografts, in NOD-SCID-Gamma mice (NSG), tumors progressed rapidly and in a highly reproducible manner through low (7 days) and high burden stages (day 14). Animals were humanely killed at or soon after day 21, due to large tumor size and/or impending ulceration.

**Co-expression of hNIS in CAR retroviral constructs**. Tri-cistronic retroviral vectors were constructed to co-express P28ζ or

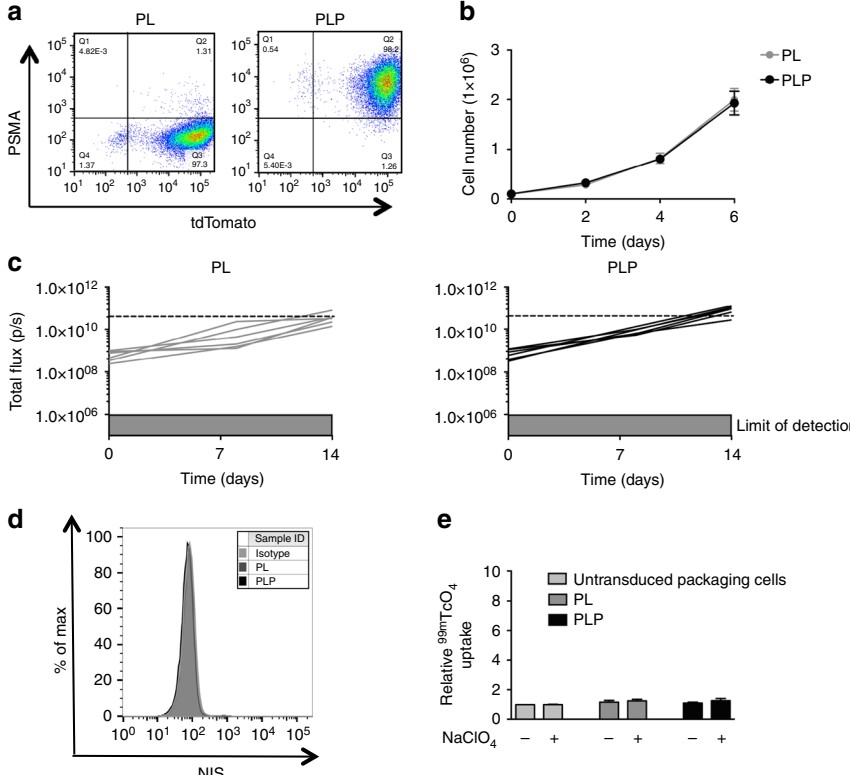

**Fig. 1** Establishing a PSMA$^{+/-}$ xenograft model. **a** Representative flow cytometry pseudo-color density plot of PL and PLP tumor cells to assess PSMA and tdTomato expression. **b** In vitro growth of PL and PLP tumor cells. **c** In vivo growth of PL and PLP subcutaneous tumor xenografts was assessed using bioluminescence imaging. Differences were not statistically significant. **d** Representative flow cytometry histograms of PL and PLP tumor cells to detect hNIS expression on the cell surface. **e** Relative in vitro uptake levels of PL and PLP tumor cells treated with 0.1 MBq $^{99m}TcO_4^-$ in the presence or absence of 10 mM NaClO$_4^-$. Empty RD packaging cells were used as a control. All graphs presents mean ± s.e.m. of three independent experiments

a matched control CAR, truncated from the intracellular domain of CD28 (PTr); an interleukin (IL)-4 responsive chimeric cytokine receptor, named 4αβ (coupling the IL-4 receptor (IL-4R) α ectodomain to the transmembrane and endodomain of IL-2/IL-15Rβ[46]); and the hNIS (giving 4P28ζN or the truncated control 4PTrN, (Fig. 2a) utilizing the Thosea Asigna (T2A) ribosomal skip peptide to achieve stoichiometric co-expression of all three transgenes[47]. Expression of all transgenes was demonstrated in retroviral packaging cell lines (Supplementary Fig. 1a) and in retrovirus-transduced T cells by flow cytometry (Fig. 2b). When surface expression of 4αβ levels on transduced T cells was quantified, variability was minimal between donors (Fig. 2c). Both CD4$^+$ and CD8$^+$ subsets of in vitro transduced T cells showed a predominately effector memory phenotype, accompanied by smaller numbers of central memory cells (Supplementary Fig. 2a, b). With the exception of one donor whose T cells were greater than 95% CD4$^+$ (Supplementary Fig. 2c-d), the CD4/CD8 ratio was consistently around 2:1 for all other donors.

**Functional validation of CAR efficacy and specificity.** Functional activity of the CAR was confirmed by demonstrating the ability of 4P28ζN$^+$, but not 4PTrN$^+$, T cells to elicit cytotoxicity (Fig. 2d) and release of interferon (IFN)-γ and tumor necrosis factor (TNF)-α (Fig. 2e), when co-cultivated with PLP but not PL tumor cells. In our in vivo model, P28ζ re-targeted CAR T cells elicited the complete eradication of low burden PLP tumors that had been established for 7 days, whereas truncated CAR T cells were inactive (Fig. 2f).

**Kinetics and impact of $^{99m}TcO_4^-$ uptake transduced T cells.** Function of hNIS was demonstrated by in vitro $^{99m}TcO_4^-$ uptake

by hNIS expressing T cells (Fig. 3a) and retroviral packaging cell lines (Supplementary Fig. 1b). Uptake above background was completely abrogated by addition of the competitive inhibitor, sodium perchlorate (NaClO$_4^-$), further substantiating the role of hNIS in this process (Fig. 3a and Supplementary Fig. 1b). Importantly, $^{99m}TcO_4^-$ uptake in the T cells did not affect viability, indicated by their ability to proliferate after exposure to isotope (Fig. 3b). Furthermore, $^{99m}TcO_4^-$ uptake was only seen in viable T cells, irrespective of transgene expression (Fig. 3c). Saturation of relative activity was not reached in vitro over a 120-min period (Fig. 3d). Increasing the concentration of $^{99m}TcO_4^-$ had minimal impact on relative levels of $^{99m}TcO_4^-$ uptake (Fig. 3e), and complete removal of radioactivity from the medium showed rapid efflux with reestablishment of a steady state within 15 min (Fig. 3f).

**In vivo validation of hNIS function in tumor free mice.** Next, we investigated if 4P28ζN$^+$ T cells could be detected in vivo by SPECT/CT imaging. Decreasing doses of 4P28ζN$^+$ T cells, from $5 \times 10^5$ to $0.15 \times 10^5$ in six increments, were injected subcutaneously in the tissue overlying the scapula in NSG mice. SPECT/CT images were acquired after intravenous $^{99m}TcO_4^-$ injection via the tail vein. Percentage of the total injected dose, corrected for thigh uptake to account for blood pooling, was calculated for each individual 4P28ζN$^+$ T-cell injection (Fig. 4a). The highest recovered dose was seen at the highest ($5.0 \times 10^5$) dose (Fig. 4b). Even at the lowest dose of 15,000 4P28ζN$^+$ T cells, the subcutaneous localization of the hNIS expressing cells could be clearly visualized (Fig. 4c) on whole animal SPECT/CT images. Delineation of the specificity of the hNIS reporter was

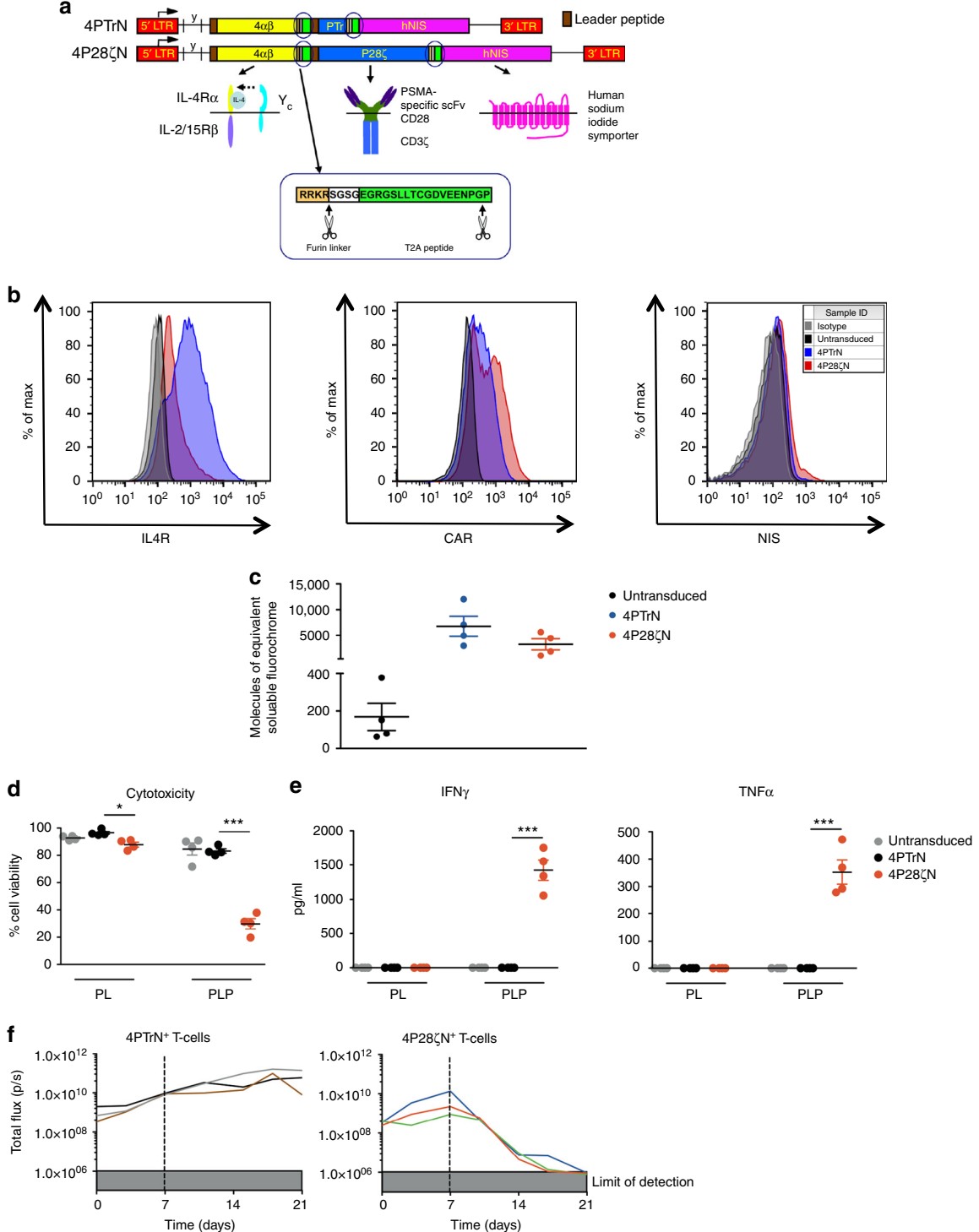

**Fig. 2** Tri-cistronic retroviral vectors function. **a** Schematic diagram of the CAR constructs. **b** Representative flow cytometry histograms of CAR, 4αβ chimeric cytokine receptor and hNIS expression on the cell surface of transduced T cells. **c** Fluorescence quantitation of cell surface expression of IL-4Rα. $n = 4$ donors/group. **d** Percentage cell viability of tumor cells 24 h after co-culture with T cells (1:1 E:T ratio) as determined by MTT assays. $n = 4$ donors/group. Graph presents mean ± s.e.m. **$P < 0.01$, ***$P < 0.001$. Supernatants from the experiments in **c** were used in ELISAs to determine the levels of (**e**) IFN-γ and TNF-α secretion by T cells. $n = 4$ donors/group. Graph presents mean ± s.e.m. ***$P < 0.001$. **f** Tumor burden of subcutaneous firefly luciferase-expressing PLP tumors treated with 4P28ζN or 4PTrN was assessed by BLI over a period of 21 days. $n = 3$ animals/group. Graph presents individual animal BLI signals

demonstrated in vivo by the evaluation of signal when administration of $^{99m}TcO_4^-$ was preceded by either PBS or sodium perchlorate ($NaClO_4^-$) in mice with high tumor burden xenografts (Fig. 4d,e). In the $NaClO_4^-$ pre-treated animals, focal areas of $^{99m}TcO_4^-$ uptake (particularly in the thyroid and stomach)

were no longer seen with the total injected dose distributed throughout the blood pool.

**CAR T-cell imaging in a low tumor burden model**. To further investigate whether migration of 4P28ζN[+] CAR T cells could be

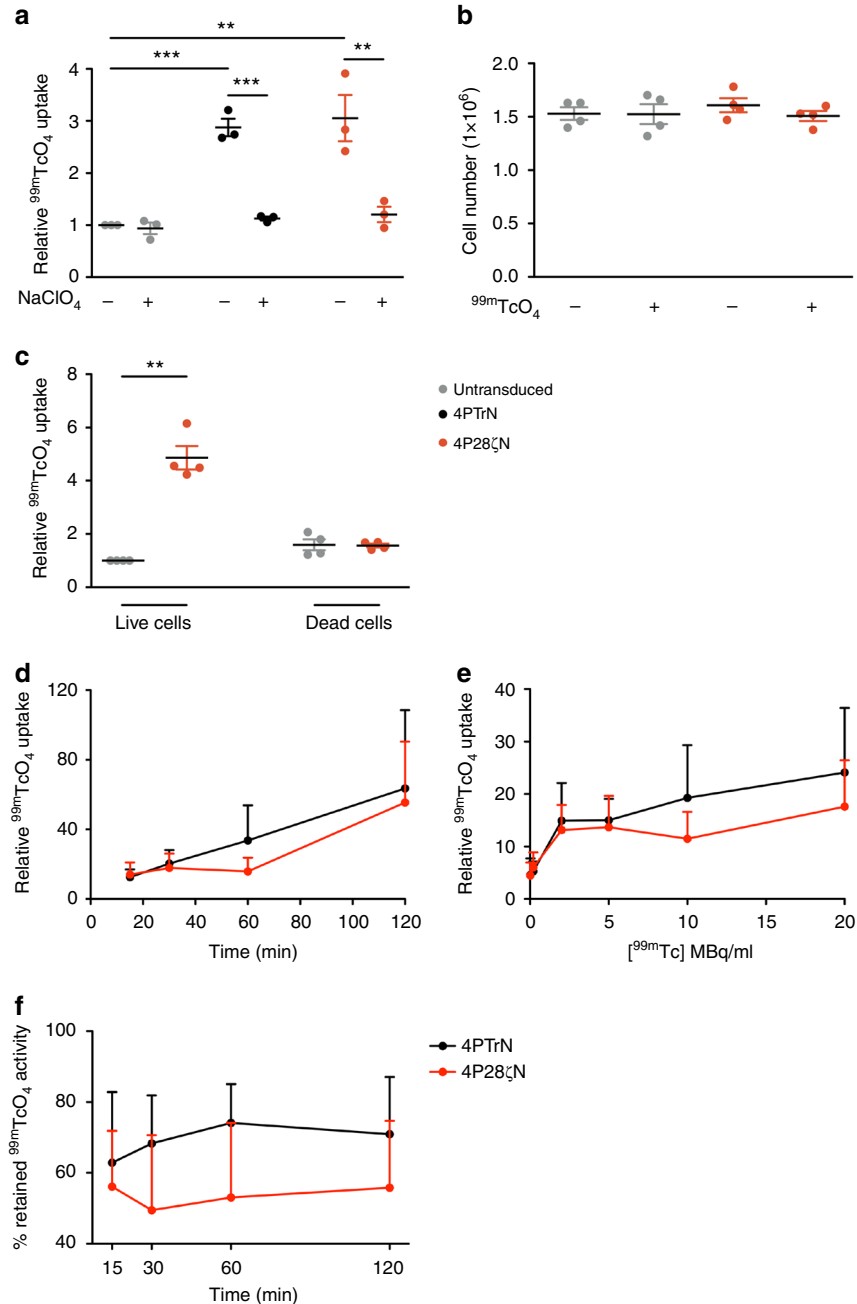

**Fig. 3** Establishing and validating hNIS function. **a** Relative in vitro uptake levels of T cells treated with $^{99m}TcO_4^-$ in the presence or absence of $NaClO_4^-$. $n$ = 3 donors/group. **b** Cell counts of untransduced or 4P28ζN T cells after 24 h culture in the presence or absence of $^{99m}TcO_4^-$. $n$ = 4 donors/group. Graph presents mean ± s.e.m. **c** Relative in vitro uptake levels in live/dead untransduced or 4P28ζN T cells treated with $^{99m}TcO_4^-$. $n$ = 4 donors/group. Graphs present mean ± s.e.m. *$P < 0.05$, **$P < 0.01$, ***$P < 0.001$. **d** Relative T-cell $^{99m}TcO_4^-$ uptake over time. $n$ = 6 donors/group. Graphs present mean + s.e.m. **e** Relative uptake of $^{99m}TcO_4^-$ by concentration. $n$ = 6 donors/group. Graphs present mean + s.e.m. **f** Efflux of $^{99m}TcO_4^-$ over time shown as % of retained activity normalized to time 0. $n$ = 3 donors/group. Graphs present mean + s.e.m.

tracked in vivo by SPECT/CT imaging, low burden (7 day) PLP tumors were established. Serial dual modality imaging was used to monitor tumor growth (BLI) and intra-tumoral accumulation of hNIS+ CAR T cells (SPECT/CT), according to the indicated experimental plan (Supplementary Fig. 3a). Baseline SPECT/CT images were acquired for normalization and showed no $^{99m}TcO_4^-$ uptake within tumors above blood pool levels (Supplementary Fig. 3b). In 4P28ζN-treated animals, tumor-derived BLI signal plateaued from the day of T-cell administration, sharply decreased from day 14 and became undetectable by day 21 (Fig. 5a and Supplementary Fig. 3c). SPECT/CT scanning

demonstrated that this therapeutic response correlated with focal intra-tumoral uptake of $^{99m}TcO_4^-$ by day 16 (day 9 post T-cell injection) in two of the three 4P28ζN animals (Fig. 5b). This activity was absent by day 21, at which time tumors had been eradicated (Fig. 5b). Tomographic images are shown in Fig. 5c. SPECT/CT imaging revealed intra-tumoral accumulation of T cells at day 21 in one mouse treated with 4PTrN+ T cells (gray line; Fig. 5c, d). While this was accompanied by a leveling off in the rate of tumor progression (indicated by BLI signal), tumor control did not occur (Fig. 5a, b) and all 4PTrN+ T-cell and PBS-treated animals were culled owing to advanced tumor size at day 21.

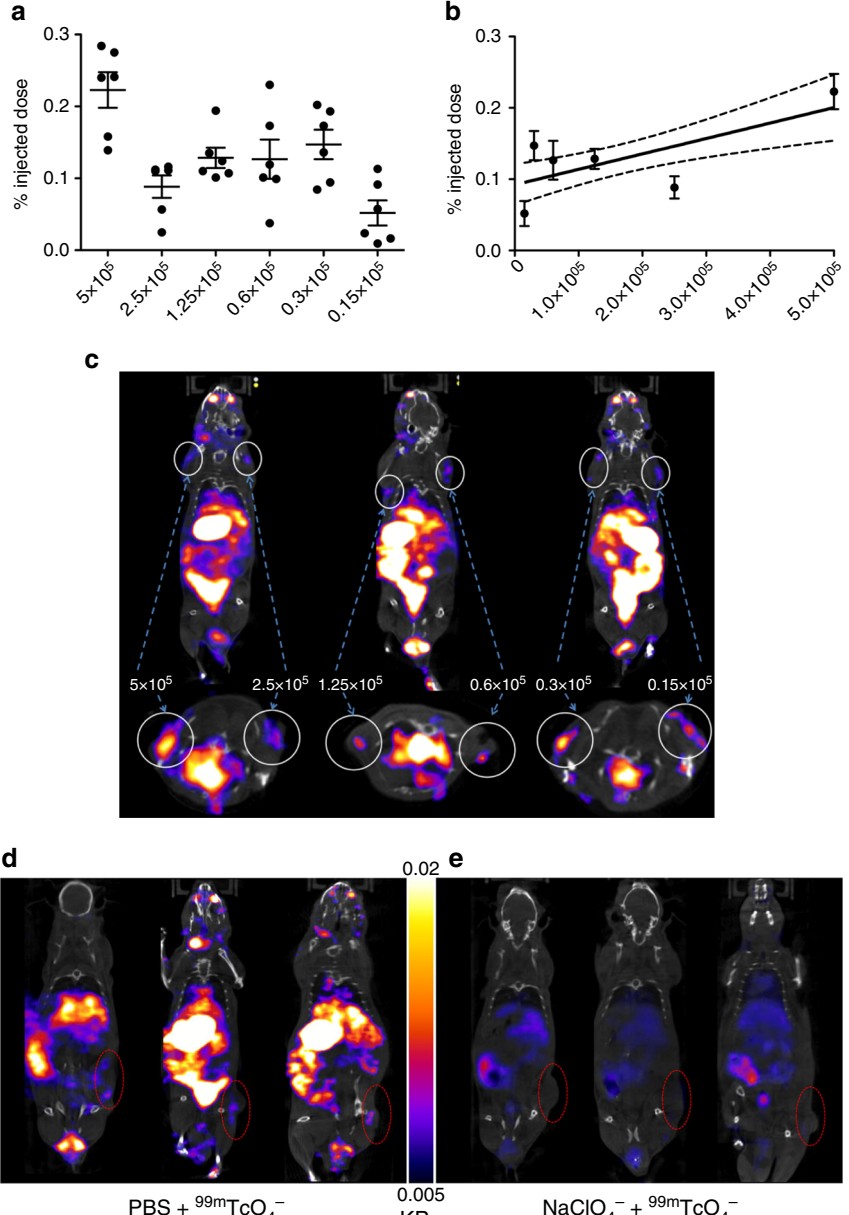

**Fig. 4** Sensitivity and specificity of $^{99m}TcO_4^-$ uptake in vivo. **a** Differing doses of 4P28ζN$^+$ T cells injected subcutaneously over the shoulders were quantified by SPECT/CT. Data was corrected to thigh to account for blood pooling. $n = 6$ animals/group, expressed as mean ± SEM. **b** Linear regression of data in **a**. Representative SPECT/CT images for all dose levels shown in **c**. Animals with established (14 days) PLP xenografts were given ACT with $1 \times 10^6$ 4P28ζN$^+$ T cells. Seven days post-ACT, animals were given i.p. injections of either (**d**) PBS or (**e**) sodium perchlorate (10 mg/ml in PBS) 30 min before infusion of 20 MBq $^{99m}TcO_4$. SPECT/CT images for all individual animals shown

Despite physiologic uptake of isotope by endogenous mouse NIS (mNIS) in stomach, salivary glands and thyroid, resolution of imaging of T-cell accumulation in the tumors was excellent, particularly in whole-body videos, where the tumor localization of CAR T cells is clear in the 4P28ζN-treated animal day 9 post T-cell infusion (Movies 1–6), and derived still images (Fig. 5d). Removal of mNIS-expressing organs was not required to visualize intra-tumoral T-cell uptake in images. A repeat 'low tumor burden' experiment was undertaken in which tumor control was not observed in 4P28ζN-treated animals (Supplementary Fig. 4a). Notably, the T-cell donor for this experiment was a significant outlier with an extremely high CD4$^+$/CD8$^+$ ratio (Supplementary Fig. 2d), which may explain the lack of efficacy. Importantly, however, marked accumulation of 4P28ζN T cells was seen in

two of the three animals, albeit at the later time point of 14 days following administration of CAR T cells (Supplementary Fig. 4b,c).

**CAR T-cell imaging in a high tumor burden model.** Next, we investigated CAR T-cell accumulation using the 'high tumor burden' model. A sub-optimal dose of $1 \times 10^6$ 4P28ζN$^+$ T cells was administered to mice bearing $14^-$ day established PL or PLP tumors (experimental protocol; Supplementary Fig. 5a). As expected, tumor control was not seen in any animal, owing to low dose of T cells and larger tumor size (Fig. 6a, Supplementary Fig. 5c and Supplementary Fig. 6a). Mean BLI signal in PLP tumors at day 14 were higher in comparison to PL tumors.

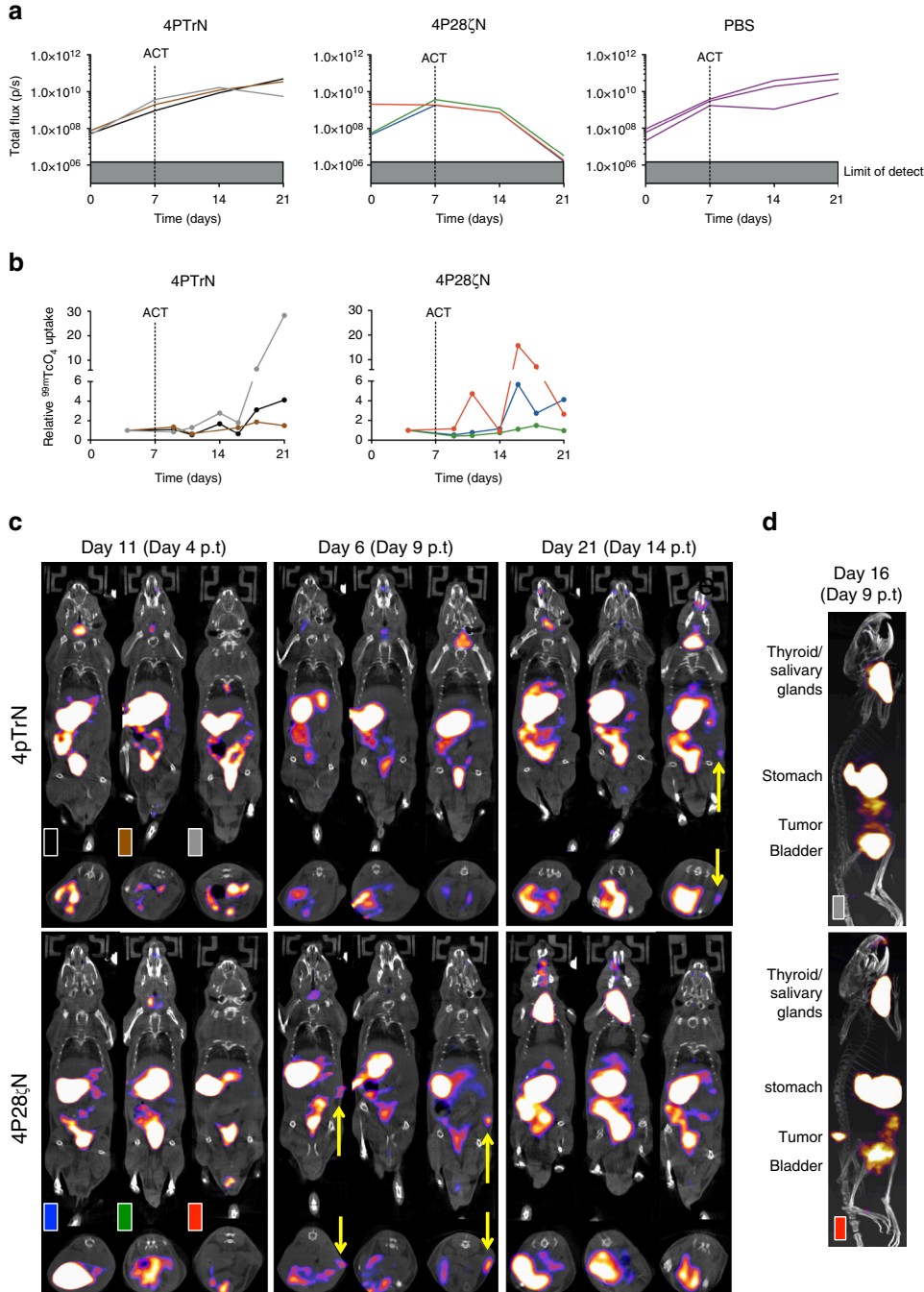

**Fig. 5** In a 'low tumor burden' model, 4P28ζN T-cell accumulation in tumors is observed through SPECT-CT imaging. **a** Burden of subcutaneous firefly luciferase-expressing PLP tumors was assessed by BLI over a period of 21 days. $n = 3$ animals/group. Graphs present individual animal BLI signals. **b** $^{99m}TcO_4^-$ activity in the tumor was quantified through 3D ROI analysis and normalized to baseline. $n = 3$ animals/group. Graph presents individual animal SPECT signals. **c** SPECT-CT images depicting the accumulation of NIS-expressing T cells in the tumors (indicated by the yellow arrows) of the individual animals (p.t = post-therapy). **d** Representative Day 9 post T-cell transfer images from a 4PTrN and 4P28ζN-treated animal with labeling of physiological $^{99m}TcO_4^-$ uptake and excretion via the urinary tract

Nonetheless, mean BLI signal in PLP animals only increased marginally in contrast to a marked increase in PL animals (Fig. 6b). This deceleration of growth is consistent with sub-therapeutic PSMA-dependent activity of 4P28ζN⁺ CAR T cells. Pre-treatment baseline SPECT/CT images again showed no additional accumulation of isotope in either xenograft, compared to blood pool levels (Supplementary Fig. 6b). Detectable homing of 4P28ζN T cells to PLP tumors occurred within 24 h of T-cell

inoculation, with progressive amplification of this signal there-after (Figs. 6c, d, 4c and Supplementary Fig. 6b,c). Indeed, two days after T-cell administration, four PLP tumors demonstrated focal T-cell accumulation but only one PL tumor (Fig. 6d and Supplementary Fig. 6c). However, by day 21, PLP and PL mice demonstrated focal $^{99m}TcO_4^-$ uptake within tumors (Fig. 6d and Supplementary Fig. 6c). Once again, the anatomical resolution of T-cell localization in the tumor was noteworthy (Fig. 6d).

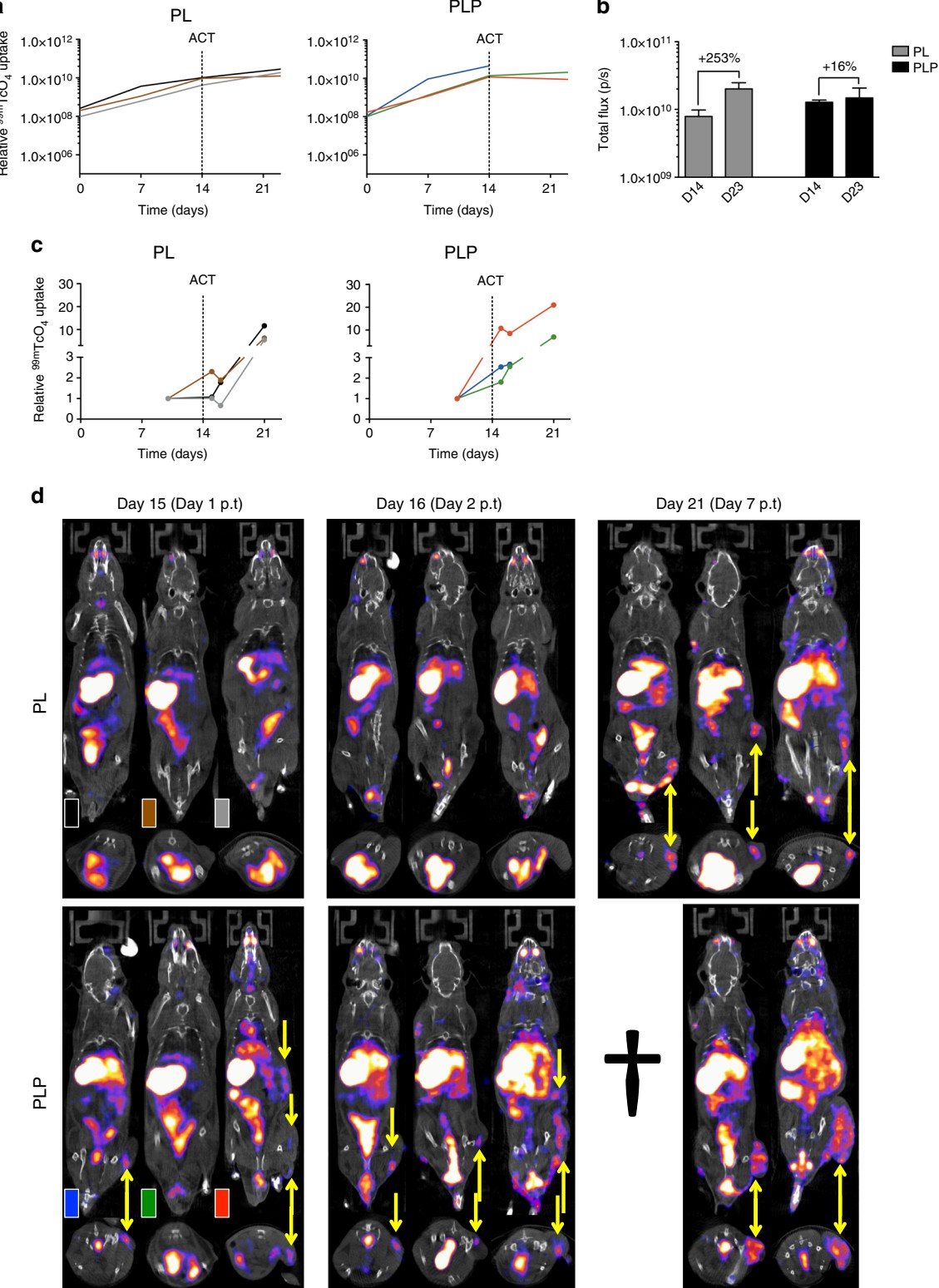

**Fig. 6** In a 'high tumor burden' model, SPECT-CT imaging shows rapid 4P28ζN T-cell accumulation within the tumor mass. **a** Burden of subcutaneous firefly luciferase-expressing PL and PLP tumors was assessed by BLI over a period of 21 days. $n = 3$ animals/group. Graphs present individual animal BLI signals. **b** Tumor growth at day 14 and day 23 as assessed by BLI. $n = 3$ animals/group. Graph presents mean + s.e.m. **c** $^{99m}TcO_4^-$ activity in the tumor was quantified through 3D ROI analysis and normalized to baseline. $n = 3$ animals/group. Graphs present individual animal SPECT signals. **d** SPECT-CT images depicting the accumulation of NIS-expressing T cells in the tumors (indicated by the yellow arrows) of the individual animals (p.t = post-therapy; $^\dagger$ = dead animal)

**Immunohistochemistry confirms CAR T-cell localization to tumor**. Finally, staining of frozen sections of 'high tumor burden' xenografts for human CD3ζ was undertaken 2 and 7 days after CAR T-cell or PBS injection. In keeping with the SPECT/CT data, infiltration of 4P28ζN$^+$ but not control 4PTrN$^+$ T cells was observed in PSMA-expressing PLP tumors (but not PL tumors) by day 2. A delayed, non-antigen specific, infiltration of 4PTrN cells in PL and PLP tumors was visible by day 7 (Supplementary Fig. 7).

## Discussion

These data demonstrate for the first time that human CAR T cells can be repeatedly imaged in vivo using a hNIS-based reporter system. This allowed us to explore the inter-relationship between intra-tumoral CAR T-cell accumulation and disease control.

Translational SPECT/CT imaging with the tracer $^{99m}TcO_4^-$ offers potential for excellent utility as a reporter system for cell therapy studies for three key reasons. Firstly, imaging with $^{99m}TcO_4^-$ is routinely undertaken in the clinic. Secondly, the approach described here is inexpensive and is compatible with both PET and SPECT/CT scanners, both of which are standard items of equipment found in clinical nuclear medicine departments. Finally, the practical and straightforward nature of this technology makes hNIS highly amenable to incorporation into clinical CAR T-cell protocols.

In small animals, SPECT/CT delivers high levels of resolution. In human imaging, SPECT offers moderately lower sensitivity and spatial resolution than PET. In practice, this is not likely to be an impediment to the use of this approach in the clinic and recent advances in SPECT imaging technology affording gains in both resolution and sensitivity are enabling quantitation with SPECT as part of the clinical routine[48]. If greater resolution/sensitivity proves necessary, it should be noted that hNIS is also compatible with a growing range of PET tracers[49], albeit at a far greater cost to implement.

Organ-specific tracer uptake by hNIS in the stomach and thyroid would limit the use of this approach to image T cells directed against tumors in those anatomical regions. One potential solution to address this entails the administration of a radio-opaque substance orally, prior to imaging, to allow activity blocking in the stomach. This could be accomplished using clinically available agents such as barium sulfate[50]. Excretion of free isotope via the kidneys, with accumulation in the bladder, is a limitation of many clinically useful radiotracers. This could mask uptake into CAR T cells in the anatomical area. One possible solution to this would be urinary catheterization, or hydration followed by urination just prior to administration of isotope, in human studies. In practice, renal excretion of F-18-fluorodeoxyglucose ($^{18}$F-FDG), a tracer widely used in the clinic for PET imaging, does not diminish the clinical utility of this imaging approach.

Early translation of novel CAR T-cell approaches in man must focus on safe delivery, efficient manufacturing and the demonstration of meaningful signals of efficacy. Some clinical studies involving CAR T-cell immunotherapy have indicated that severe and occasionally fatal toxicity can result from this complex but highly promising therapeutic strategy[10,16,51]. Cytokine release syndrome (CRS) is a prevalent toxic event and in fact may represent a crude biomarker of sustained tumor rejection. Neurological toxicity is also emerging as a challenge in the CD19 CAR T-cell protocols. The mechanism of this organ-specific toxicity remains incompletely understood at the present time. In developing CAR T-cell immunotherapeutic approaches for solid malignancies, identification of tumor-specific targets or targets that are only found on 'expendable' normal tissues (such as CD19

on B cells) remains a pressing challenge. Consequently, on-target, off-tumor toxicity represents a major concern in the development of these experimental therapies. Clinical experience indicates that such events can be unanticipated prior to first in man evaluation, making careful clinical trial design and execution all the more critical[15,16]. A further complexity in CAR T-cell clinical development is the role of conditioning regimens that achieve lymphodepletion and thereby potentiate efficacy of the CAR T-cell product. These regimens can elicit significant toxic effects in patients, which can be challenging to differentiate from adverse events that are attributable to the CAR T cells themselves. The ability to rapidly and non-invasively determine the direct contribution of the CAR T cells to an adverse event will empower clinicians with a greater mechanistic understanding of these increasing clinical occurrences. Moreover, such information would enable the evidence-based deployment of targeted approaches to toxicity reversal, thereby improving the safety of early phase CAR T-cell development. In the event of long-term persistence of CAR T cells, the system described here would also render delayed toxic events amenable to investigation using a straightforward imaging-based approach.

Together, these data demonstrate the utility of hNIS-based CAR T-cell imaging at the translational interface. While use of hNIS is limited by physiologic expression in the thyroid, salivary glands and stomach, it is not expressed in hematopoietic cells, which represents a key strength. Improved understanding of clinically relevant anti-tumor activity is also likely to be gained by implementation of the proposed approach in man. Notably, we observed an oscillating pattern of T-cell accumulation in some mice in which tumor rejection occurred. This mimics the pattern that has previously been described when bulky tumors are rejected in a T-cell receptor-dependent manner[52]. The technology is unencumbered allowing clinical investigators freedom to incorporate this system into the development of novel CAR T-cell platforms. In short, the incorporation of hNIS into CAR-encoding vector systems provides a non-invasive platform to investigate the serial bio-distribution of these cells in patients. This offers an unparalleled opportunity to correlate the anatomical location and number of CAR T cells throughout the body with the occurrence of toxicity and the dynamic alterations that occur within malignant deposits.

## Methods

**Recombinant DNA constructs**. All recombinant DNA constructs were expressed via the retroviral vector SFG. An optimized furin cleavage site (RRKR), SGSG linker and T2A peptide[47] were utilized to ensure stoichiometric expression of more than one construct in the same open reading frame. Where necessary sequences were 'wobbled' maximally to reduce the risk of vector instability. The J591 antibody sequence (used to construct the PSMA-specific P28ζ CAR) was provided by Dr Neil Bander (Cornell University, NY). The P28ζ CAR was co-expressed with the 4αβ chimeric cytokine receptor and hNIS (named 4P28ζN). A truncated CAR, in which the signaling domains of CD28 and CD3ζ were removed (named 4PTrN), was similarly co-expressed with 4αβ and hNIS. Human PSMA was synthesized as an NcoI/XhoI fragment and cloned into SFG using those restriction sites. SFG-Luc-RFP was generated as previously described[53], required fragments for cloning were synthesized by Genscript (Piscataway, NJ).

**Cell lines**. The 293VEC-RD114 retroviral packaging cell line was provided by Dr. Manuel Caruso, Biovec Pharma, Québec, Canada. The PC3-LN3 cell line was provided by Prof. Sue Eccles, Institute of Cancer Research, London. The H29 transient retroviral producer cell line was obtained from Dr. Michel Sadelain, Memorial Sloan Kettering Cancer Center, USA.

All cell lines were propagated in DMEM (Life Technologies) supplemented with 10% FBS (Sigma-Aldrich). In addition, the media used for the H29 cell line was supplemented with 1 mg puromycin (Thermo Fisher Scientific), 165 mg geneticin (Thermo Fisher Scientific), and 1 mg tetracycline (Sigma-Aldrich), which were removed prior to transfection.

All cell lines were screened regularly for mycoplasma contamination.

**Retroviral transduction of cells**. Plasmids were transfected into the H29 transient retroviral producer cell line using polyethylenemine (PEI; Sigma-Aldrich). Briefly, PEI and plasmid DNA were mixed at a 3:1 ratio in serum-free DMEM and incubated for 15 min at room temperature. The mixture was then added to H29 cells, which were incubated at 37 °C, 5% $CO_2$. Viral supernatants were collected 72 h post-transfection and were used to generate stable retroviral producing 293VEC-RD114 cell lines as well as PSMA-expressing PC3-LN3 cells.

Human peripheral blood mononuclear cells were obtained from healthy volunteers, after written informed consent, under approval of the Guy's and St Thomas' Research Ethics Committee (reference 09/H0804/92). PBMCs were isolated by low-density centrifugation on Ficoll Pacque (Sigma-Aldrich), activated for 48 h with CD3/CD28-coated paramagnetic Dynabeads at a 3:1 bead:cell ratio (Thermo Fisher Scientific), transduced by centrifugation on Retronectin (Takara) coated plates, and cultured in 100 IU/ml IL-2 or 30 ng/ml IL-4 (Miltenyi Biotec). Media and cytokines were replenished every 2 days. Transduced T cells were propagated in RPMI 1640 (Life Technologies) supplemented with 10% AB serum (Sigma-Aldrich)

All media were supplemented with 2 mmol/L L-glutamine (Life Technologies), 100 units/mL penicillin, and 100 μg/mL streptomycin (Life Technologies).

**T-cell co-cultures**. PC3-LN3 cell lines were seeded at $1 \times 10^5$ cells/well in 24-well plates and incubated overnight at 37 °C, 5% $CO_2$. T cells were added to the monolayers at a 1:1 ratio (with no added cytokines) and the co-cultures were incubated for 24 h at 37 °C, 5% $CO_2$. Supernatants were collected at this point for cytokine detection. T cells were aspirated and residual monolayers were carefully washed with PBS. To quantify residual viable tumor cells, wells were incubated with 250 μg 3-(4,5-dimethylthiazol-2-yl)-2,5-diphenyltetrazolium bromide (MTT; Sigma-Aldrich) in media for 2 h at 37 °C, 5% $CO_2$. Formed purple formazan crystals were then re-suspended in 300 μl DMSO. Absorbance was measured at 560 nm, and tumor cell viability was calculated ((absorbance of monolayer and T-cell co-culture/absorbance of tumor monolayer alone)×100).

**Enzyme-linked immunosorbent assays**. Cytokine concentrations were measured in T-cell culture supernatants by using Human IFN-gamma DuoSet ELISA and Human TNF-alpha DuoSet ELISA (R&D Systems, Inc.), according to the manufacturer's instructions.

**Cell proliferation assays**. PC3-LN3 cell lines were seeded at $1 \times 10^5$ cells/well in six-well plates and incubated at 37 °C, 5% $CO_2$. At the indicated times, cells were trypsinized and counted with a hemocytometer using the Trypan Blue exclusion method.

**In vitro $^{99m}$Tc-pertechnetate assays**. $^{99m}$TcO$_4^-$ used in all experiments was provided by the Radiopharmacy, Nuclear Medicine Department, Barts Health NHS Trust, London. The $^{99m}$TcO$_4^-$ was eluted with saline from a generator, which had been eluted in the last 24 h to minimize the build-up of $^{99}$Tc.

T cells at $1 \times 10^6$ cells/condition were treated with either PBS or 10 mM sodium perchlorate (Sigma) and incubated for 30 min at 37 °C, 5% $CO_2$. Cells were then incubated with 0.1 MBq $^{99m}$TcO$_4^-$ for 30 min at 37 °C, 5% $CO_2$. Cells were washed three times in ice-cold PBS before quantifying the remaining radioactivity using a gamma counter (LKB Wallac 1282 Compugamma CS).

The viability of $^{99m}$TcO$_4^-$-treated T cells was determined by cell counts with a hemocytometer using the Trypan Blue exclusion method. T cells were seeded at $1 \times 10^6$ cells/well in six-well plates and were treated with either PBS or 0.1 MBq $^{99m}$TcO$_4^-$ and were incubated for 24 h (with no added cytokines) at 37 °C, 5% $CO_2$ after which viability was determined.

Specific $^{99m}$TcO$_4^-$ uptake in viable cells was determined by measuring the radioactivity of the cells using a gamma counter. T cells at $1 \times 10^6$ cells/condition were either left untreated or fixed with 4% PFA for 15 min on ice, which was then followed by incubation with 0.1 MBq $^{99m}$TcO$_4^-$ for 30 min at 37 °C, 5% $CO_2$. Cells were washed three times in ice-cold PBS before quantifying the remaining radioactivity.

To determine $^{99m}$TcO$_4^-$ uptake at various concentrations, various activities of $^{99m}$TcO$_4^-$ in 50 μl PBS were added to 450 μl of $5 \times 10^5$ transduced T cells kept at 37 °C in PBS in triplicate to give activity concentrations of 0.02, 0.2, 2, 5, 10 and 20 MBq/ml and incubated for 1 h (inverting every 5–10 min). Cells were centrifuged and washed in ice-cold PBS twice before quantifying the cell-associated radioactivity using a gamma counter. To determine $^{99m}$TcO$_4^-$ uptake over time, $5 \times 10^5$ transduced T cells were cultured with 2 MBq/ml $^{99m}$TcO$_4^-$ (0.5 ml total volume) and treated as above at 15, 30, 60 or 120 min in triplicate before counting in a gamma counter. For efflux experiments $5 \times 10^5$ transduced T cells were cultured in triplicate with 2 MBq/ml $^{99m}$TcO$_4^-$ for 1 h. Cells were centrifuged and washed twice in ice-cold PBS then the radioactivity of the cells measured using a gamma counter (time point 0) or resuspended in pre-warmed PBS and cultured for 15, 30, 60 or 120 min before centrifuging and washing twice with ice-cold PBS. Cell-associated radioactivity was measured using a gamma counter.

**Flow cytometry**. Flow cytometry was performed on a BD-LSRII cytometer and data analyzed with FlowJo software (TreeStar). Expression of the P28ζ CAR was detected using the goat anti-mouse F(ab')2 IgG RPE-conjugate at a concentration of 10 μg/ml (Dako, 13416.1), which binds to murine sequences within the J591-derived single chain antibody fragment. The following primary antibodies were used: PSMA was detected using a mouse anti-human PSMA at a concentration of 10 μg/ml (MBL Life Science, K0142-3), a mouse anti-human CD124 at a concentration of 5 μg/ml (BD Biosciences, 552178) was used to detect the 4αβ chimeric cytokine receptor, and a mouse anti-human NIS at a 1:50 dilution (Imanis Life Sciences, REA003) was used to detect NIS on the cell surface. For immunophenotyping and determining the T-cell ratio, the following antibodies were used: CD4$^-$FITC (eBiosciences, 11-0048-41), CD4$^-$APC (eBiosciences, 17-0048-41), CD8-FITC (eBiosciences, 11-0089-42), CD45RO-PE (eBiosciences, 12-0457-42) and CCR7-APC (eBiosciences, 17-1979-42) all at a concentration of 0.25 μg/test. For fluorescence quantitation of number of CAR constructs on the cell surface, the Quantum R-PE MESF (Bangs Laboratories, Inc) kit was used as per the manufacturer's instructions. The following secondary antibodies were used: goat anti-mouse F(ab')2 IgG RPE-conjugate at a concentration of 10 μg/ml (Dako, 13416.1) and Affinipure goat anti-mouse IgG (Fcγ fragment specific) Alexa Fluor 647-conjugate at a 1:100 dilution (Jackson ImmunoResearch Laboratories Inc, 115-605-008). Positive staining was assessed by comparison to the relevant isotype control antibody and/or FMO.

**Xenograft model and adoptive CAR T-cell therapy**. Male NOD scid gamma (NSG (JAX strain, NOD.Cg-Prkdc$^{scid}$ll2rg$^{tm1Wjl}$/SzJ)) mice were used, age 6 weeks to 8 weeks old (Charles River UK). This study was performed strictly adhering to the UK Home Office guidelines and licenses governing this work. Mice were inoculated with $2.5 \times 10^5$ tumor cells in the right flank. At the indicated time points, mice received $1 \times 10^6$ CAR$^+$ T cells intravenously. T cells were used at 12–14 days post-transduction.

**Immunohistochemistry**. Subcutaneous xenografts were established in NSG mice, as described above, for 14 days. The mice then received $1 \times 10^6$ CAR$^+$ T cells intravenously. T cells were used at 12–14 days post-transduction. Tumors were collected at the indicated time points and snap frozen in optimum cutting temperature (O.C.T) embedding matrix (Cellpath$^{TM}$). Blocks were sectioned and stored at −80 °C. Slides were thawed at room temperature for 5 min, fixed in 10% formalin for 15 min and washed in PBS. Slides were incubated with the primary antibody, monoclonal rabbit anti-human CD3ζ (clone MRQ-39, OriGene), at a 1/1000 dilution with DAKO diluent (Ely, Cambridgeshire, UK), for 1 h. Secondary staining with the VECTASTAIN Elite ABC-HCP kit was undertaken according to manufacturers instructions (Burlingame, California, USA). Slides were counterstained with Gills no 3 Haematoxylin, differentiated in 1% acid alcohol, dehydrated and mounted.

**In vivo blocking of $^{99m}$TcO$_4^-$ uptake**. Subcutaneous xenografts were established in NSG mice, as described above, for 14 days. The mice then received $1 \times 10^6$ CAR$^+$ T cells intravenously. T cells were used at 12–14 days post-transduction. At day 7 post T-cell transfer and 30 min before $^{99m}$TcO$_4^-$ infusion, the mice received either an i.p injection of 500 μl PBS or sodium perchlorate (10 mg/ml in PBS). NSG mice were then injected via the tail vein with 20 MBq of $^{99m}$TcO$_4^-$ 1 h prior to SPECT imaging, which was performed using the methods described below.

**Threshold determination in vivo**. NSG mice were inoculated in the subcutaneous space overlying the scapula with indicated doses of CAR$^+$ T cells. T cells were used at 12–14 days post-transduction. The mice were then subjected to SPECT/CT imaging 1 h post T-cell inoculation, using the method described below.

**In vivo Nano-SPECT/CT imaging**. NSG mice were injected via the tail vein with 20 MBq of $^{99m}$TcO$_4^-$ 1 h prior to SPECT imaging. The mice were then anaesthetized with 2% isoflurane gas and 1 L/min oxygen and whole-body SPECT images obtained using a NanoSPECT/CT four-head camera (Bioscan, Inc., Washington DC, USA) fitted with 1.4 mm pinhole collimators in helical scanning mode (24 projections, 40 min acquisition). CT images were acquired with a 45-kVP X-ray source (1500 ms exposure, 180 projections). A multiple bed system was used to allow three mice to be imaged simultaneously and body temperature was maintained via a warm air heating system. CT reconstruction was undertaken with InVivoScope (inviCRO LLC, Boston, USA) and SPECT reconstruction with HiSPECT, (Scivis GmbH, Göttingen, Germany) then merged and region of interest analysis performed using VivoQuant software (inviCRO LLC, Boston, USA).

$^{99m}$Tc-pertechnetate uptake in the tumor was quantified with 3D ROI analysis. The percentage injected dose was calculated (with decay correction), and normalized to baseline SPECT values.

**In vivo bioluminescence imaging**. Bioluminescence imaging (BLI) was performed at the indicated time points using an IVIS Lumina III in vivo imaging system (PerkinElmer). Analysis was performed using Living Image 4.3.1 software (Caliper Life Sciences). Mice underwent intraperitoneal injection with D-luciferin (150 mg/kg; PerkinElmer) and were imaged under isoflurane anesthesia after 10 min. Image acquisition was conducted on a 20 cm field of view with medium binning for a

range of exposure times (0.2–2 min). Animals were inspected daily and killed when experimental endpoints had been achieved or when symptomatic due to tumor progression. Tumor burden, as determined by IVIS imaging, was assessed using region of interest (ROI) analysis. The resulting signal summations (total flux in units of photons/s) were plotted for each mouse.

**Statistical analyses**. All figures are representative of at least three independent experiments unless otherwise noted. All graphs report mean ± s.e.m values of biological replicates. Statistical significance of the data was calculated using a Student's $t$-test. Significance is designated with an asterisk in all figures ($*P < 0.05$, $**P < 0.01$, and $***P < 0.001$).

**Data availability**. The authors declare that all the data supporting the findings of this study are available within the article and its supplementary information files and from the corresponding author upon reasonable request.

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

## Acknowledgements

This work was supported by the Medical Research Council, Guy's and St Thomas Charity, Worldwide Cancer Research, Prostate Cancer UK, The Academy of Medical Sciences, the Experimental Cancer Medicine Centre at King's College London, the King's Health Partners/ King's College London Cancer Research UK Cancer Centre and by the National Institute for Health Research (NIHR) Biomedical Research Centre based at Guy's and St Thomas' NHS Foundation Trust and King's College London. We would like to acknowledge Breast Cancer Now for support in immunohistochemistry image capture and analysis. The views expressed are those of the authors and not necessarily those of the NHS, the NIHR or the Department of Health.

## Author contribution

N.E.-S., R.K. and S.P. designed and performed experiments, analyzed results and wrote the manuscript, J.S. co-supervised the work, designed experiments, analyzed results and contributed insight to the manuscript, C.C. performed experiments. P.G. analyzed results, J.F. designed experiments, analyzed results and contributed insight to the manuscript, J.M. designed experiments and wrote the manuscript.

## Additional information

**Competing interests:** J.M. is chief scientific officer of Leucid Bio, which is a spinout company focused on development of cellular therapeutic agents. The remaining authors declare no competing interests.

