## [Peer Review File · Nature Communications]

Reviewers' comments:

Reviewer #1 (Remarks to the Author):

This manuscript describes use of the human NIS as an imaging reporter gene for tracking CAR T-cells in vivo. The manuscript is reasonably well written and well organized. However, there are several concerns with the study.

7) Page 5, top: "The symporter co-transport two sodium ions with one iodide ion, generating a sodium gradient to rapidly concentrate inorganic iodide within the thyroid gland", is a misstatement. The sodium gradient exists a priori, generated by the action of the sodium-potassium ATPase pump of the plasma membrane. The NIS simply uses the electrochemical potential of the sodium gradient as the energy source to drive co-transport of the iodide in a concentrative manner into the cell. Please restate this passage.

X) Page 9, mid. Despite statements to the contrary, the widely distributed background uptake of the tracer and the many physiological sites of uptake would interfere with the interpretation of images clinically, especially when the location of the T-cell target (primary tumor, mets, nodes) is not known a priori.

X) Page 9, bottom. Figure 5C. average BLI signal in PLP animals appeared to increase marginally, not decrease as stated. Please check this.

X) Page 10, mid. "... demonstrate for the first time that CAR T-cells can be imaged..." is a bit of hyperbole given the many reports on use of this reporter strategy in the literature for many types of cell trafficking studies.

6) Figure 3 E-F. The images appear mixed up in the figure.

x) Figures 4 and 5. It would seem that n=3 for many of these experiments is not robust. This reviewer would like to see n=5 to 6 animals for each to increase the rigor of the studies.

X) Figure 5E. Again, this is not convincing. If one had not known the location of the tumor, the background and physiological uptake of $^{99m}\text{TcO}_4$ in the various organs and sites would make it extremely difficult to identify cellular infiltration prospectively with this approach.

X) Figure 5. In vivo blocking experiments should be performed with cold perrhenate or iodide to prove specificity of the signal in vivo.

X) Figure 5. More validation of the imaging signal is required. How did imaging signal in vivo correlate with the analysis of T-cell infiltrate within the tumor by IHC? What was the comparative gold standard for cell infiltrate? More analysis is needed to prove that the imaging assay properly reported cell trafficking. What was the threshold of cell infiltrate detection? How did cell number correlate with signal? What was the slope of the correlation curve?

Reviewer #2 (Remarks to the Author):

Emami-Shahri et al. demonstrate the potential utility of using a human reporter gene, hNIS, to serially image human CAR T cells in mice using broadly available tracers and imaging platforms. This study addresses an unmet need with strong translational potential. Whereas hNIS is not novel as a non-invasive imaging resource, its use in T cells could find broad applications. The data establish the feasibility and specificity of hNIS imaging outside of areas with background. The

sensitivity of the method, however, is unclear.

Major criticism

A better quantification of the method's sensitivity is necessary. This could be accomplished by autoradiography or histology to confirm uptake in hNIS-CAR T cells and enumerate tumor-infiltrating lymphocytes. SPECT quantification is difficult and not done in clinical routine.

The discussion is rather poor, not addressing the strengths and weakness of the approach:

- in vitro uptake assay only at 1 timepoint (30 min) + only one amount of radioactivity (0.1 MBq ^{99m}Tc) – measurement of different amounts of activity and different time-points?
- Interaction of hNIS (as reporter gene) and $^{99m}\text{TcO}_4^-$ (probe): Kinetics of uptake? What happens with this probe once it locates on the reporter gene? Is it taken up in cells? How fast? How long is it retained?
- ^{99m}Tc is not organified - which is also a positive aspect- but it is subject to rapid efflux? Is this seen in in vitro uptake assays done at multiple time-points? And would this possibly limit its sensitivity? Please briefly comment (the authors highlight its clinical utility).
- Clinical translation - Thyroid blockade? Blockade on CAR T cells as well?
- Metabolism is well studied but physiologic excretion in urine limits evaluation of urinary tract and urinary bladder (and pelvis)
- Known threshold limits uptake of NIS mediated radio-iodine uptake (Vadysirisack et al.) – in vitro uptake studies to show plateau of uptake?
- Difficult quantification of hNIS-cell surface expression – otherwise could be done: receptor expression quantification – comment?

Other points:

Methods

- Cell line authentication?
- List of abbreviations missing
- Method for radiotracer preparation – kit?
- Maybe include phenotyping (CD4/CD8 counts)?

Figures

- Figure 1: in A the axes (markers) are not readable; also in D too small
- Figure 2: B: axes (markers) are not readable
- Fig. 2 C: labeling missing for "PLP"
- Figure 3: E and F are mislabeled
- Fig. 3: E and F: only 1 mouse each? Or 1 control mouse and 2 mice with CAR T cells? But only 1 mouse shown? Dose/amount injected? Time-point imaged? Autoradiography and histology? (And 1×10^7 are a lot of T cells... or rather 1×10^6 as mentioned in methods)
- Fig. 4: uptake only in 2 out of 3 mice – why? Could there be issues with the post-translational modification/glycosylation and membrane-transport/expression of hNIS? Autoradiography/histochemical evaluation?

Reviewer #3 (Remarks to the Author):

This study is the first example of the use of the sodium-iodide symporter (NIS) reporter gene for imaging CAR T cells. This is an important topic as the use of CAR T cells is increasing dramatically, and CAR T cells are generally not able to be tracked in the clinical setting after they've been administered, which hampers the ability to assess the efficacy and potential toxicities from the cells.

1. In Figures 4 and 5 tracer uptake ($^{99m}\text{TcO}_4$) is seen in PSMA positive tumors (PLP) and PSMA negative tumors (PL) using CAR T cells with a functional PSMA-specific scFv (4P28ζN) and with a non-functional scFv (4PTrN). The authors assume that the uptake at the tumor sites is representative of the presence of CAR T cells, and while CAR T cells have been shown to traffic to a tumor site in the absence of specific targeting, this uptake still needs to be validated in some fashion. It is possible that the uptake that is seen is due to nonspecific retention of tracer in the tumors, which are larger at later time points and thus have more nonspecific tracer uptake relative to the baseline scan. Thus, it would be good to confirm that the uptake at the tumor sites is specific for the presence of CAR T cells by either doing a blocking study (with excess cold iodide) or by using CAR T cells that don't contain functional NIS. Also, since the mice are being sacrificed at day 21 anyways (or soon thereafter), the tumor sites could be removed and the presence of CAR T cells could be assayed, and compared with the imaging data.

2. In Figure 4C there seem to be 2 mice treated with 4PTrN T cells that have tracer uptake: the grey line and black line. But in the text (on page 9) it states that there was only 1 mouse with tracer uptake. Why was the mouse represented by the black line not included?

3. The caption for Figure 4 lists A-D but the figure uses A-E. This should be corrected.

4. In Figure 5D it looks like the mice with the highest uptake on day 2 include the mice represented by the red, blue, brown, and green lines. Thus, it appears that one of the PL mice (brown line) has uptake on day 2. But in the text (on page 10) it states that uptake could not be visualized in any of the PL tumors. Why was the PL mouse represented by the brown line not mentioned?

5. Figure 2C: the "PLP" label on the x-axis got cut off and needs to be fixed.

6. Figure 3C: the caption should note that cells were incubated for 24h .

7. Figure 3E: this figure is mislabeled as "Ut", it should be labeled as "4P28ζN". Also, the number of cells that were injected should be specified in the caption. Was the limit of detection ever assessed by injecting fewer cells?

8. Figure 3F: this figure is mislabeled as "4P28ζN", it should be labeled as "Ut". Also, the number of cells that were injected should be specified in the caption.

9. Figure 3G: this figure is a bit confusing because it shows two bars with the same "4P28ζN" label (presumably from two different mice). This data might look better with a single "4P28ζN" bar that shows the mean +/- s.e.m.

10. On page 4 it states: "this approach is critically limited by the internalization of the SSTR2 receptor on interaction with its substrates, leading to poor resolution of imaging." It's not clear how internalization of the SSTR2 receptor would result in poor image resolution. Do you mean that the sensitivity is reduced? Also, "critically limited" seems a bit strong; internalization of the receptor is a limitation, but maybe not a critical limitation.

Response to Reviewers' comments:

Reviewers' comments:

Reviewer #1(Remarks to the Author):

This manuscript describes use of the human NIS as an imaging reporter gene for tracking CAR T-cells in vivo. The manuscript is reasonably well written and well organized. However, there are several concerns with the study.

1) Page 5, top: “The symporter co-transport two sodium ions with one iodide ion, generating a sodium gradient to rapidly concentrate inorganic iodide within the thyroid gland”, is a misstatement. The sodium gradient exists a priori, generated by the action of the sodium-potassium ATPase pump of the plasma membrane. The NIS simply uses the electrochemical potential of the sodium gradient as the energy source to drive co-transport of the iodide in a concentrative manner into the cell. Please restate this passage.

Response: We thank the reviewer for this correction. Text has been amended to read: “The symporter co-transport two sodium ions with one iodide ion, capitalizing on the electrochemical potential of the sodium gradient generated by the sodium-potassium ATPase (Na^+/K^+ -ATPase) pump of the plasma membrane, to rapidly concentrate inorganic iodide within the thyroid gland.”

2) Page 9, mid. Despite statements to the contrary, the widely distributed background uptake of the tracer and the many physiological sites of uptake would interfere with the interpretation of images clinically, especially when the location of the T-cell target (primary tumor, mets, nodes) is not known a priori.

Response: We agree with the reviewer on this point and have amended text on page 15 to read “Use of hNIS is limited by physiologic expression in the thyroid, salivary glands and stomach, which may potentially compromise tumor imaging at those sites. Importantly however, it is not expressed in hematopoietic cells, which represents a key strength.” Furthermore, on page 11 in support of the reviewers’ statement we concluded that “Despite physiologic uptake of isotope by endogenous mouse NIS (mNIS) in stomach, salivary glands and thyroid, resolution of imaging of T-cell accumulation in the tumors was excellent, particularly in whole body videos (**Supplementary Figure 4**) and derived still images (**Figure 5D**).”

3) Page 9, bottom. Figure 5C. average BLI signal in PLP animals appeared to increase marginally, not decrease as stated. Please check this.

Response: We have amended this statement. Text on page 11 now reads: “Mean BLI signal in PLP tumors at day 14 were higher in comparison to PL tumors. Nonetheless, mean BLI signal in PLP animals only increased marginally in contrast to a marked increase in PL animals (**Figure 6B**). This deceleration of growth is consistent with sub-therapeutic PSMA-dependent activity of 4P28ζN⁺ CAR T-cells.”

4) Page 10, mid. “... demonstrate for the first time that CAR T-cells can be imaged...” is a bit of hyperbole given the many reports on use of this reporter strategy in the literature for many types of cell trafficking studies.

Response: The first sentence of the Discussion has been amended to reflect this critique, stating “While the hNIS reporter has been used in a range of cell tracking studies, these data demonstrate for the first time that human CAR T-cells can be repeatedly imaged *in vivo* using this system.”

5) Figure 3 E-F. The images appear mixed up in the figure.

Response: Our apologies for this mix up in the original manuscript. These images have been removed altogether and replaced with threshold data (**Figure 4A and B**). These data demonstrate that CAR-hNIS⁺ T-cells can be imaged effectively following the administration of as few as 1.5 x 10⁴ cells.

6) Figures 4 and 5. It would seem that n=3 for many of these experiments is not robust. This reviewer would like to see n=5 to 6 animals for each to increase the rigor of the studies.

Response: This point is well made. We have increased the numbers to n=6/group for the ‘low tumor burden model’ and n=6/group for the ‘high tumor burden model’, with an additional three ‘high tumor burden model’ animals imaged in the blocking experiment. This takes the total number of imaged CAR T-cell treated animals demonstrated to 33 (**Figure 5 and 6, Supplementary Figure 5 and 7 and Figure 4C**).

7) Figure 5E. Again, this is not convincing. If one had not known the location of the tumor, the background and physiological uptake of 99m-TcO₄ in the various organs and sites would make it extremely difficult to identify cellular infiltration prospectively with this approach.

Response: These data are now in Figure 5D: With experience of reviewing images generated following the administration of NIS transported isotopes, the physiologic distribution of tracer is highly consistent and uptake in other areas is clear to see. The increased n numbers for the imaging experiments show this. We have also added the statement: “Despite physiologic uptake of isotope by endogenous mouse NIS (mNIS) in stomach, salivary glands and thyroid, resolution of imaging of T cell accumulation in the tumors was excellent, particularly in whole body videos (**Supplementary Figure 4**) and derived still images (**Figure 5D**).

8) Figure 5. In vivo blocking experiments should be performed with cold perrhenate or iodide to prove specificity of the signal in vivo.

Response: Blocking experiments have been carried out where by animals are injected with PBS or sodium perchlorate followed by $^{99m}\text{TcO}_4^-$. These data show the abrogation of uptake in the tumors despite the relative increase in circulating $^{99m}\text{TcO}_4^-$ as the isotope is no longer being taken up in the thyroid/stomach etc (**Figure 4C**).

9) Figure 5. More validation of the imaging signal is required. How did imaging signal *in vivo* correlate with the analysis of T-cell infiltrate within the tumor by IHC? What was the comparative gold standard for cell infiltrate? More analysis is needed to prove that the imaging assay properly reported cell trafficking. What was the threshold of cell infiltrate detection? How did cell number correlate with signal? What was the slope of the correlation curve?

Response: We have added new data to demonstrate the excellent sensitivity of this approach with a threshold of signal down to 1.5×10^4 CAR T cells in ROI (**Figure 4A and B**). Immunohistochemistry for CD3z shows infiltration of CD3⁺ corresponding the groups in which tumor uptake signal is detected *in vivo* (**Supplementary Figure 8**)

Reviewer #2 (Remarks to the Author):

Emami-Shahri et al. demonstrate the potential utility of using a human reporter gene, hNIS, to serially image human CAR T cells in mice using broadly available tracers and imaging platforms. This study addresses an unmet need with strong translational potential. Whereas hNIS is not novel as a non-invasive imaging resource, its use in T cells could find broad applications. The data establish the feasibility and specificity of hNIS imaging outside of areas with background. The sensitivity of the method, however, is unclear.

Major criticism

A better quantification of the method's sensitivity is necessary. This could be accomplished by autoradiography or histology to confirm uptake in hNIS-CAR T cells and enumerate tumor-infiltrating lymphocytes. SPECT quantification is difficult and not done in clinical routine.

The discussion is rather poor, not addressing the strengths and weakness of the approach:

Response: We have expanded the discussion and added a section on page 13-14 on the anatomical weaknesses of the approach as follows: "Organ-specific tracer uptake by hNIS in the stomach and thyroid would limit the use of this approach to image T-cells directed against tumors in those anatomical regions. One potential solution to address this entails the administration of a radio-opaque substance orally, prior to imaging, to allow activity blocking in the stomach. This could be accomplished using clinically available agents such as barium sulfate⁵⁰. Excretion of free isotope via the kidneys, with accumulation in the bladder, is a limitation of many clinically useful radiotracers. This could mask uptake into CAR T-cells in the anatomical area. One possible solution to this would be urinary catheterization, or hydration followed by urination just prior to administration of isotope, in human studies. In practice,

renal excretion of F-18-fluorodeoxyglucose (^{18}F -FDG), a tracer widely used in the clinic for PET imaging, does not diminish the clinical utility of this imaging approach.”

- in vitro uptake assay only at 1 timepoint (30 min) + only one amount of radioactivity (0.1 MBq $^{99\text{m}}\text{Tc}$)
– measurement of different amounts of activity and different time-points?

Response: We have expanded the *in vitro* uptake and kinetics data to show multiple time points and concentration variables (**Figure 3 D and E**). These data show saturation of uptake over 5MBq $^{99\text{m}}\text{TcO}_4$ and variability of uptake after 60min.

- Interaction of hNIS(as reporter gene) and $^{99\text{m}}\text{TcO}_4$ - (probe): Kinetics of uptake? What happens with this probe once it locates on the reporter gene? Is it taken up in cells? How fast? How long is it retained?

Response: Efflux has been demonstrated with new data (**Figure 3F**). This shows that efflux reaches steady state within the first 15 minutes of culture. Quantitation is the subject of a multidisciplinary grant application (Physics, biology and mathematic) This is a complex body of work which seeks to generate an algorithm for image quantitation. This important work will form a stand-alone manuscript for publication.

- $^{99\text{m}}\text{Tc}$ is not organified - which is also a positive aspect- but it is subject to rapid efflux? Is this seen in in vitro uptake assays done at multiple time-points? And would this possibly limit its sensitivity? Please briefly comment (the authors highlights its clinical utility).

Response: Efflux, over 5 time points, has been demonstrated with new data (**Figure 3F**). This shows that efflux reaches steady state within the first 15 minutes of culture.

- Clinical translation - Thyroid blockade? Blockade on CAR T cells as well?

Response: We have added the following text to the discussion on page 13-14: ‘One potential solution to address this entails the administration of a radio-opaque substance orally, prior to imaging, to allow activity blocking in the stomach. This could be accomplished using clinically available agents such as barium sulfate.’ Blocking *in vivo* would be challenging, as blockade is irreversible and would block hNIS function on the CAR T cells also.

- Metabolism is well studied but physiologic excretion in urine limits evaluation of urinary tract and urinary bladder (and pelvis)

Response: We have added the following text to the discussion on page 14: ‘Excretion of free isotope via the kidneys, with accumulation in the bladder, is a limitation of many clinically useful radiotracers. This could mask uptake into CAR T-cells in the anatomical area. One possible solution to this would be urinary catheterization, or hydration followed by urination just prior to administration of isotope, in human studies. In practice, renal excretion of F-18-fluorodeoxyglucose (^{18}F -FDG), a tracer widely used in the clinic for PET imaging, does not diminish the clinical utility of this imaging approach.”

- Known threshold limits uptake of NIS mediated radio-iodine uptake (Vadysirisack et al.) – in vitro uptake studies to show plateau of uptake?

Response: We have expanded the *in vitro* uptake and kinetics data to show multiple time points and concentration variables (**Figure 3 D and E**). These data show saturation of uptake over 5MBq $^{99m}\text{TcO}_4^-$ and variability of uptake after 60min. We have also explored threshold *in vivo* demonstrating signal down to 0.15×10^5 CAR T cells

- Difficult quantification of hNIS-cell surface expression – otherwise could be done: receptor expression quantification – comment?

Response: We agree with the reviewer that quantification of hNIS at the cell surface is challenging due to limited reagents that bind the extracellular domain of the symporter. We plan to address this is our future work looking a detailed quantification of hNIS reporter gene imaging, using compartmental modelling and the development of an algorithm in collaboration with mathematicians.

Other points:

Methods

- Cell line authentication?

Response: We have STR profiling from 22 FEB 2015 for the tumour cells lines. Packaging cell lines (293 Vec imparting RD114 envelope, were provided under MTA with STR profiling in place).

- List of abbreviations missing.

Response: We apologise for omitting this. We have now added a list to page 2.

- Method for radiotracer preparation – kit?

Response: We have added the following wording to the methods on page 19 to describe the elution procedure for isotope: $^{99m}\text{TcO}_4^-$ used in all experiments was provided by the Radiopharmacy, Nuclear Medicine Department, Barts Health NHS Trust, London. The $^{99m}\text{TcO}_4^-$ was eluted with saline from a generator, which had been eluted in the last 24 hours to minimise the build-up of ^{99}Tc .

- Maybe include phenotyping (CD4/CD8 counts)?

Response: We have added phenotyping and CD4/CD8 ratios (**Supplementary Figure 2**)

Figures

- Figure 1: in A the axes (markers) are not readable; also in D too small

Response: We have been through the figures in this resubmission carefully to ensure all axes are readable.

- Figure 2: B: axes (markers) are not readable

Response: We have been through the figures in this resubmission carefully to ensure all axes are readable.

- Fig. 2 C: labeling missing for “PLP”

Response: We apologise for this error, which has now been resolved in the Figure.

- Figure 3: E and F are mislabelled

Response: These data have been removed and are now replaced with threshold date (**Figure 4A and B**).

- Fig. 3: E and F: only 1 mouse each? Or 1 control mouse and 2 mice with CAR T cells? But only 1 mouse shown? Dose/amount injected? Time-point imaged? Autoradiography and histology? (And 1 x 10e7 are a lot of T cells... or rather 1 x 10e6 as mentioned in methods)

Response: These data have been removed and are now replaced with threshold date (**Figure 4A and B**).

- Fig. 4: uptake only in 2 out of 3 mice – why? Could there be issues with the post-translational modification/glycosylation and membrane-transport/expression of hNIS? Autoradiography/histochemical evaluation?

Response: This is an interesting thought. Due to Home Office limits that govern animal work undertaken in the UK, we are limited as to the number of procedures we can carry out/week, necessitating careful selection of time points for imaging. It is likely that, in some animals, we are missing the point at which the T cell signal is maximal in the tumors. The increase in the number of animals that have undergone T cell imaging to a total of 33 provides a clear indication of this variability but also shows that imaging of hNIS expressing CAR T cells is robust and repeatable.

Reviewer #3 (Remarks to the Author):

This study is the first example of the use of the sodium-iodide symporter (NIS) reporter gene for imaging CAR T cells. This is an important topic as the use of CAR T cells is increasing dramatically, and CAR T cells are generally not able to be tracked in the clinical setting after they've been administered, which hampers the ability to assess the efficacy and potential toxicities from the cells.

1. In Figures 4 and 5 tracer uptake ($^{99m}\text{TcO}_4$) is seen in PSMA positive tumors (PLP) and PSMA negative tumors (PL) using CAR T cells with a functional PSMA-specific scFv (4P28ζN) and with a non-functional scFv (4PTrN). The authors assume that the uptake at the tumor sites is representative of the presence of CAR T cells, and while CAR T cells have been shown to traffic to a tumor site in the

absence of specific targeting, this uptake still needs to be validated in some fashion. It is possible that the uptake that is seen is due to nonspecific retention of tracer in the tumors, which are larger at later time points and thus have more nonspecific tracer uptake relative to the baseline scan. Thus, it would be good to confirm that the uptake at the tumor sites is specific for the presence of CAR T cells by either doing a blocking study (with excess cold iodide) or by using CAR T cells that don't contain functional NIS. Also, since the mice are being sacrificed at day 21 anyways (or soon thereafter), the tumor sites could be removed and the presence of CAR T cells could be assayed, and compared with the imaging data.

Response: Blocking experiments have been carried out whereby animals are injected with PBS or sodium perchlorate followed by $^{99m}\text{TcO}_4^-$. These data show the abrogation of uptake in the tumors despite the relative increase in circulating $^{99m}\text{TcO}_4^-$ as the isotope is no longer being taken up in the thyroid/stomach etc (**Figure 4C**). Immunohistochemistry for CD3z shows infiltration of CD3⁺ cells corresponding to the groups in which tumor uptake signal is detected *in vivo* (**Supplementary Figure 8**).

2. In Figure 4C there seem to be 2 mice treated with 4PTrN T cells that have tracer uptake: the grey line and black line. But in the text (on page 9) it states that there was only 1 mouse with tracer uptake. Why was the mouse represented by the black line not included?

Response: At the imaging time point, the uptake was not visible on the still image of the ROI. We have since increased n numbers for *in vivo* studies to a total of n = 33 imaged animals (**Figure 5, Figure 6 and Supplementary Figures 5 and 7**).

3. The caption for Figure 4 lists A-D but the figure uses A-E. This should be corrected.

Response: We apologise for this error. This figure (now **Figure 5** in the new manuscript) has been re-ordered and corrected.

4. In Figure 5D it looks like the mice with the highest uptake on day 2 include the mice represented by the red, blue, brown, and green lines. Thus, it appears that one of the PL mice (brown line) has uptake on day 2. But in the text (on page 10) it states that uptake could not be visualized in any of the PL tumors. Why was the PL mouse represented by the brown line not mentioned?

Response: At the imaging time point the uptake was not visible on the still image of the ROI. We have since increased n numbers for *in vivo* studies to a total of n = 33 imaged animals (**Figure 5, Figure 6 and Supplementary Figures 5 and 7**).

5. Figure 2C: the "PLP" label on the x-axis got cut off and needs to be fixed.

Response: Apologies, this has been resolved in the Figure.

6. Figure 3C: the caption should note that cells were incubated for 24h.

Response: We have amended the legend to the Figure to read as follows: '(B) Cell counts of untransduced or 4P28ζN T cells after 24-hour culture in the presence or absence of $^{99m}\text{TcO}_4^-$. n = 4

donors/group.'

7. Figure 3E: this figure is mislabeled as "Ut", it should be labeled as "4P28ζN". Also, the number of cells that were injected should be specified in the caption. Was the limit of detection ever assessed by injecting fewer cells?

Response: These data have been removed completely from our re-submission and replaced with new data to demonstrate the excellent sensitivity of this approach with a threshold of signal down to 1.5×10^4 CAR T cells in ROI (**Figure 4A and B**).

8. Figure 3F: this figure is mislabeled as "4P28ζN", it should be labeled as "Ut". Also, the number of cells that were injected should be specified in the caption.

Response: These data have been removed completely from our re-submission and replaced with new data to demonstrate the excellent sensitivity of this approach with a threshold of signal down to 1.5×10^4 CAR T cells in ROI (**Figure 4A and B**).

9. Figure 3G: this figure is a bit confusing because it shows two bars with the same "4P28ζN" label (presumably from two different mice). This data might look better with a single "4P28ζN" bar that shows the mean +/- s.e.m.

Response: These data have been removed completely from our re-submission and replaced with new data to demonstrate the excellent sensitivity of this approach with a threshold of signal down to 1.5×10^4 CAR T cells in ROI (**Figure 4A and B**).

10. On page 4 it states: "this approach is critically limited by the internalization of the SSTR2 receptor on interaction with its substrates, leading to poor resolution of imaging." It's not clear how internalization of the SSTR2 receptor would result in poor image resolution. Do you mean that the sensitivity is reduced? Also, "critically limited" seems a bit strong; internalization of the receptor is a limitation, but maybe not a critical limitation.

Response: This passage has been replaced, on page 5 of the revised manuscript, with: 'However, this approach has two limitations. Firstly, the SSTR2 receptor internalizes on interaction with its substrates, risking poor sensitivity, especially at lower reporter gene expressing cell density³². Secondly, SSTR2 is expressed on T-cells and other immune cells³³, accounting for the ability of octreotide analogues and their radiolabeled derivatives to inhibit T-cell function³⁴. This is clearly undesirable for a broadly applicable strategy to image therapeutic T-cell products in cancer patients.'

Reviewers' comments:

Reviewer #1 (Remarks to the Author):

Several control experiments have been performed improving the rigour and presentation.

Reviewer #2:

Editorial Note: Reviewer#2 only provided confidential comments to the editor. They found most of their concerns addressed except for what they believe was the most important point i.e. a better quantification of the method's sensitivity. However they acknowledge a succinct answer was provided in response to the other reviewers with figure 4AB and Supplementary Figure8.

Reviewer #3 (Remarks to the Author):

The authors have addressed the questions and comments well. However, new figures and text have been added, which require comment:

Figure 4A: For Figure 4A the data suggests that 5×10^5 cells is detectable, with other lower doses of subcutaneous cells at background levels of tracer uptake. To support this, there is no dose response relationship, and the uptake from 2.5×10^5 cells is nearly equivalent to the uptake from 0.15×10^5 cells. Thus, the statement that "even at the lowest dose of 15,000 4P28ζN+ T cells, the subcutaneous localization of the hNIS expressing cells could be quantified (Figure 4A) and clearly visualized (Figure 4B) on whole animal SPECT/CT images" is likely incorrect and should be modified. In addition, measurement of background activity is needed for comparison; in previous publications, background measurements were recorded from a ROI in the thigh (see Ref #29).

Figure 4D: For Figure 4D the SPECT images are windowed differently compared to the SPECT images in 4B and 4C. Presumably this was done in order to emphasize the lack of uptake in the tumors after blocking with NaClO₄. These figures should all be windowed similarly, and the uptake in the tumor after blocking with NaClO₄ should be quantified and compared with the uptake in the PBS treated mice (from Figure 4C). The statement that "No signal was detected in the tumors" of NaClO₄ treated mice should be rephrased.

Figure 5B: It would be good to clarify which graph represents the 4PTrN data and which graph represents the 4P28ζN data.

Figure 6C: It would be good to clarify which graph represents the PL tumors and which graph represents the PLP tumors.

Page 12: "Once again, the anatomical resolution of T-cell localization in the tumor was noteworthy (Figure 5D)." Should this be Figure 6D instead of 5D?

Please find our response to the reviewer's further comments below.

Reviewers' comments:

Reviewer #1 (Remarks to the Author):

Several control experiments have been performed improving the rigour and presentation.

Reviewer #2 :

Editorial note: Reviewer#2 only provided confidential comments to the editor. They found most of their concerns addressed except for what they believe was the most important point i.e. a better quantification of the method's sensitivity. However they acknowledge a succinct answer was provided in response to the other reviewers with figure 4AB and Supplementary Figure 8.

Reviewer #3 (Remarks to the Author):

The authors have addressed the questions and comments well. However, new figures and text have been added, which require comment:

Figure 4A: For Figure 4A the data suggests that 5×10^5 cells is detectable, with other lower doses of subcutaneous cells at background levels of tracer uptake. To support this, there is no dose response relationship, and the uptake from 2.5×10^5 cells is nearly equivalent to the uptake from 0.15×10^5 cells. Thus, the statement that "even at the lowest dose of 15,000 4P28ζN+ T cells, the subcutaneous localization of the hNIS expressing cells could be quantified (Figure 4A) and clearly visualized (Figure 4B) on whole animal SPECT/CT images" is likely incorrect and should be modified. In addition, measurement of background activity is needed for comparison; in previous publications, background measurements were recorded from a ROI in the thigh (see Ref #29).

Response: The data has been reanalysed and thigh uptake subtracted from each measurement. We have expressed the data as dot plots +/- SEM and also fitted a linear regression curve to illustrate the relationship between % injected dose and 4P28ζN+ T cell number. We have amended the text at the end of page 9 to read as follows: Next, we investigated if 4P28ζN+ T-cells could be detected *in vivo* by SPECT/CT imaging. Decreasing doses of 4P28ζN+ T-cells, from 5×10^5 to 0.15×10^5 in six increments, were injected subcutaneously in the tissue overlying the scapula in NSG mice. SPECT/CT images were acquired after intravenous $^{99m}\text{TcO}_4^-$ injection via the tail vein. Percentage of the total injected dose, corrected for thigh uptake to account for blood pooling, was calculated for each individual 4P28ζN+ T-cell injection (Figure 4A). The highest recovered dose was seen at the highest (5.0×10^5) dose (Figure 4B). Even at the lowest dose of 15,000 4P28ζN+ T cells, the subcutaneous

localization of the hNIS expressing cells could be clearly visualized (Figure 4C) on whole animal SPECT/CT images.

Figure 4D: For Figure 4D the SPECT images are windowed differently compared to the SPECT images in 4B and 4C. Presumably this was done in order to emphasize the lack of uptake in the tumors after blocking with NaClO₄. These figures should all be windowed similarly, and the uptake in the tumor after blocking with NaClO₄ should be quantified and compared with the uptake in the PBS treated mice (from Figure 4C). The statement that “No signal was detected in the tumors” of NaClO₄ treated mice should be rephrased.

Response: We apologise for the lack of clarity here. The images in Figure 4 D and E were all analysed and windowed identically. We have added a scale to ensure this is clear. We have also altered the text at the start of page 10 to read as follows:

Delineation of the specificity of the hNIS reporter was demonstrated *in vivo* by the evaluation of signal when administration of ^{99m}TcO₄⁻ was preceded by either PBS or sodium perchlorate (NaClO₄⁻) in mice with high tumor burden xenografts (Figure 4D and 4E). In the NaClO₄⁻ pre-treated animals, focal areas of ^{99m}TcO₄⁻ uptake (particularly in the thyroid and stomach) were no longer seen, with the total injected dose distributed throughout the blood pool.

Figure 5B: It would be good to clarify which graph represents the 4PTrN data and which graph represents the 4P28ζN data.

Response: We have added clearer labels to the figure

Figure 6C: It would be good to clarify which graph represents the PL tumors and which graph represents the PLP tumors.

Response: We have added clearer labels to the figure

Page 12: “Once again, the anatomical resolution of T-cell localization in the tumor was noteworthy (Figure 5D).” Should this be Figure 6D instead of 5D?

Response: Thank you for noticing this error. It has been amended to read, correctly, Figure 6D

REVIEWERS' COMMENTS:

Reviewer #3 (Remarks to the Author):

The authors have addressed all of the comments and questions.